# Is Land Fragmentation Facilitating or Obstructing Adoption of Climate Adaptation Measures in Ethiopia?

**Tesfaye C. Cholo [1,2,*], Luuk Fleskens [1] 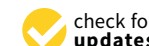, Diana Sietz [1,3] and Jack Peerlings [4]**

[1]  Soil Physics and Land Management Group, Wageningen University and Research, 6708 PB Wageningen, The Netherlands; luuk.fleskens@wur.nl (L.F.); sietz@pik-potsdam.de (D.S.)

[2]  Department of Development Economics, Ethiopian Civil Service University, P.O. Box 5648, Addis Ababa, Ethiopia

[3]  Earth System Analysis, Research Domain 1, Potsdam Institute for Climate Impact Research, Member of the Leibniz Association, P.O. Box 60 12 03, D-14412 Potsdam, Germany

[4]  Agricultural Economics and Rural Policy Group, Wageningen University and Research, 6706 KN Wageningen, The Netherlands; jack.peerlings@wur.nl

*   Correspondence: tesfaye.cholo@wur.nl

**Abstract:** Land fragmentation is high and increasing in the Gamo Highlands of southwest Ethiopia. We postulate that this substantial land fragmentation is obstructing the adoption of sustainable land management practices as climate adaptation measures. To explore this, a mixed method study was conducted with emphasis on a multivariate probit model. The results indicate that farmers adapt to climate change and variability they perceive. According to the probit model, there is no clear answer to the question whether land fragmentation facilitates or obstructs adoption of sustainable land management practices. Yet, a qualitative analysis found that farmers perceive land fragmentation as an obstacle to land improvement as adaptation strategy. Moreover, farmers invest more in land improvement on plots close to their homestead than in remote plots. However, the higher land fragmentation also promoted crop diversification, manure application and terracing. Although exogenous to farmers, we therefore suggest that land fragmentation can be deployed in climate change adaptation planning. This can be done through voluntary assembling of small neighboring plots in clusters of different microclimates to encourage investment in remote fields and to collectively optimize the benefits of fragmentation to adaptation.

**Keywords:** land fragmentation; sustainable land management; adaptation; Gamo Highlands

## 1. Introduction

Agriculture in developing countries is vulnerable to the increasing impacts of climate change, which are exacerbating food and livelihood insecurities [1,2]. Adaptation to climate change is the only strategy available for farmers. Adaptation is defined as "adjustment in natural or human systems in response to actual or expected climatic stimuli or their effects, which moderates harm or exploits beneficial opportunities" [3] (p. 869). Farmers in developing countries apply different adaptation measures [4,5]. Some of these adaptation measures fall into the category of sustainable land management (SLM) practices. SLM is defined as "a knowledge-based procedure that helps integrate land, water, biodiversity, and environmental management (including input and output externalities) to meet rising food and fiber demands while sustaining ecosystem services and livelihoods" [6]. SLM practices that serve as adaptation measure include combining crops and trees to adapt climate change, and using soil and water conservation measures such as terracing and irrigation [7–9].

Adaptive SLM practices also comprise of crop diversification [4,5] and the use of irrigation and new crop varieties [8,9]. SLM, as a strategy to adapt to climate change and variability, can result in increased food productivity and better output [10,11], although the results vary according to the specific practice used and the agro-climatic conditions [12]. Consequently, the use of SLM can build resilience to climate change [7,13].

The main barriers to adaptation in Africa are lack of access to technology, information, credit and incentives [8,9,14]. Moreover, the use and effectiveness of SLM practices as a strategy for adaptation to climate change and variability are affected by land fragmentation (LF) for several reasons. First, LF means higher commuting costs [15,16], which lead to efficiency losses [17,18]. Second, LF can potentially increase border disputes and lead to loss of land for plot demarcation purposes and access routes [19]. Third, LF restricts investment in land in the form of soil fertility enhancement and erosion control [15,20,21] and rangeland fragmentation results in rangeland degradation in southern Ethiopia [22]. Fourth, LF decreases the number of alternative uses of remote plots, for example, remote plots are not used to plant 'demanding' crops, i.e., crops that require intensive care [15,23]. These reasons imply that LF that involves the spatial disintegration of plots into separate and multiple parcels [24] limits the choice of climate adaptation measures.

However, LF may also enhance the use of SLM practices to adapt to climate change and variability. Plots scattered across diversified agro-ecological zones enhance crop diversification [25,26], smoothening labor requirements throughout the year [26–28] and freeing up time to invest in SLM. Moreover, heterogeneous scattered plots spread (climate-related) risk of production failure [26,27,29] and the soil fertility of arable land is positively affected by LF [30].

The Gamo Highlands is a mountainous area in southwest Ethiopia where there is significant LF. Subsistence farmers in this area are exposed to the adverse impacts of increasing climate change [31,32]. The effect of LF on adoption of SLM as a strategy to deal with the adverse effects of climate change has not yet been studied in this region. LF is (largely) exogenous to farmers, i.e., it is not an adaptation strategy. There are three reasons for this. First, land is inherited by farmers, which is something they cannot influence. Second, land redistribution programs have not been implemented in the Gamo Highlands unlike in the North of Ethiopia (Tigray and Amhara). Finally, farmers only have land use rights in Ethiopia, they cannot lease or sell land, implying that farmers do not have opportunities to speculate and transfer land use rights. However, we cannot exclude the possibility that farmers do exchange land use rights but we expect this to be rare; one reason could be that because of off-farm employment, some farmers have an incentive to reduce the amount of land they farm.

The purpose of this paper is to analyze the conditions under which LF is either facilitating or obstructing climate change adaptation in subsistence agriculture in the Gamo Highlands and to inform the debate on the advantages and disadvantages of LF by providing additional empirical evidence.

The next section of the paper describes the study area, data and methodology; this is followed by a section on results and discussion. A final section concludes with whether LF promotes or hinders the adoption of SLM measures to adapt to a changing climate.

## 2. Materials and Methods

### 2.1. Study Area

This study was carried out in Dita district in the Gamo Highlands. According to Dita Woreda Statistics Bureau (DWSB), the district lies at an elevation of between 1250 and 3568 m above sea level and is located at 6°12′19″ N, 37°36′37″ E [33]. The area has a long agricultural history and is largely dominated by smallholder subsistence agriculture. The district has a high population density and high topography. Due to its rugged character, the land covers two distinct agro-ecological zones (AEZs) with different opportunities for cropping; the highland area constitutes 60% and the midland area 40%. The annual rainfall of the district is bimodal and is between 2500 mm and 3500 mm. The temperature ranges between 10 °C and 23 °C.

According to [15], LF in South Asia is caused by population growth, customary land division among sons, and lack of land markets and off-farm employment opportunities. These factors also play a role in the Gamo Highlands and will inevitably lead to an increase in LF in the future. In the north-western highlands, farmers manage four to five parcels of land on average [20], and in the south-central part of Ethiopia, farms are fragmented into two to three parcels [34]. However, in the Gamo Highlands, the average number of plots per farm is 21. Projected climate change is expected to hold back agricultural progress in Ethiopia if no adaptation practices are adopted [35]. Farmers in the Gamo Highlands recognize the existence of climate change and use various measures to cope with climate change and variability. However, lack of access to credit and climate change information constrain their efforts to adapt [8,9].

### 2.2. Study Set-Up and Data

An econometric technique was used to analyze whether plot dispersion hinders or promotes the adoption of SLM as an adaptive strategy to climate change. A qualitative method was also used to explain quantitative results by examining farmers' opinions on how fragmentation influences the use of adaptation measures.

For this, both primary and secondary data were collected. Primary data was collected using a household survey. Three 'kebeles' (the lowest level administrative unit in Ethiopia) were selected from 24 kebeles in Dita district, because the local government's geographic information system (GIS) data inventory had been fully completed for these 3 kebeles. A stratified random sampling technique was used to select farmers from the kebeles that were home to 1009 households. The number of plots per farm and the AEZs that these plots were in were used to form 10 strata. Then, using a simple random sampling approach, 300 households were selected proportionately from each stratum. From each household, the household head was interviewed. Quantitative data was collected from 297 households using a questionnaire. To complement the quantitative analysis, qualitative information was collected through three focus group discussions, one in each kebele. Each focus group discussion involved 12 persons comprising farmers, experts and extension workers. In addition, key-informant interviews were held with 9 persons comprising farmers and experts. GIS data collected by experts employed by the local government using Global Positioning System (GPS) devices to register land use rights were used as secondary data to assess plot scattering.

### 2.3. Model Specification

For the quantitative analysis, this study uses a multivariate probit (MVP) model. A MVP model is an extension of a probit model used to jointly estimate multiple equations with correlated error terms [36–39] and applied when the assumption of independence of irrelevant alternatives holds [36]. The MVP model is particularly suitable when the decision to adopt a particular measure is expected to be conditional on the adoption of another complementary measure (direct relationship between the error terms of adoption equations) or may be influenced by other available substitutes (negative correlation). In contrast, the application of a binary probit, or logit model in this case, ignores the correlation in error terms of adoption equations and would lead to biased or inefficient estimation [21,40,41].

The dependent variable of the MVP model reflects the choice of *m* distinct SLM practices and is given by:

$$y^*_{im} = \beta x_{im} + \gamma LFI_{im} + \theta w_{im} + \varepsilon_{im} \quad i = 1, \ldots, I \ \& \ m = 1, \ldots, M \tag{1}$$

where $y^*_{im}$ is the latent dependent variable, which represents the level of household $i'$s net benefits from adaptation, and the distinct binary choice of a household is given by:

$$y_{im} = \begin{cases} 1, \ if \ y^*_{im} > 0 \\ 0, \ if \ y^*_{im} \leq 0 \end{cases} \tag{2}$$

Specifically, $y_{im}$ denotes the *m*th observed response of the *i*th household; $y_{im}$ equals 1 if the farmer *i* adopts SLM practice *m* and zero otherwise. Vectors of explanatory variables that are expected to determine $y_{im}$ comprise the socioeconomic characteristics of households, $x_{im}$ LF indicators, $LFI_{im}$ and $w_{im}$ denoting farmer *i*'s perception about climate variability and its impacts over the last 25 years. Error terms $\varepsilon_{im}$ are distributed as multivariate normal, each with a mean of zero and a variance covariance matrix *v*, where *v* has a value of one on the leading diagonal and correlations $\rho_{kj} = \rho_{jk}$ as off-diagonal elements [42]. In our case, $\rho_{jk}$ is the correlation coefficient of the error terms corresponding to any two adaptation equations to be estimated in the model. A positive value of $\rho_{jk}$ denotes that any two measures used are complementary while a negative correlation is interpreted as measures used being substitutes [41,43]. Stata software version 13.1 was used for model estimation.

## 3. Results

This section describes the dependent variables and explains the measurement of explanatory variables. It also presents and discusses the results of the study. Dependent variables were identified as a subset of 13 surveyed SLM practices used as climate adaptation measures, i.e., only those SLM practices that could be associated with perceived climate change and variability were included; the latter is hence presented first.

### 3.1. Descriptive Statistics

#### 3.1.1. Climate Change and Variability Perception

Farmers were asked about their perception of climate change and variability for the long rainy season (called the '*meher*') over the last 25 years. Most farmers perceived temperature as increasing (90%) and precipitation as decreasing (84%) (see Table 1). For the same period, 62% of farmers perceived soil erosion impacts to be decreasing and 58% perceived crop productivity to be increasing. Even though we lacked measured climatic data for the study area, farmers' perceptions of long-term climate change and variability were consistent with meteorological data [8] and with previous studies in Ethiopia and Sub-Saharan Africa [35,44–46].

**Table 1.** Summary statistics of variables.

| Variable | Description | % | Mean | SD | Min | Max |
|---|---|---|---|---|---|---|
| **SLM practices** | | | | | | |
| seed | 1 if a farmer alters quality seed use | 76.1 | | | | |
| manure | 1 if a farmer alters manure use | 93.9 | | | | |
| indigenous | 1 if a farmer alters indigenous tree planting | 73.7 | | | | |
| terrace | 1 if a farmer alters stone terrace or soil bund | 83.8 | | | | |
| diversification | 1 if a farmer alters crop diversification | 84.5 | | | | |
| enset | 1 if a farmer alters planting more enset | 84.5 | | | | |
| legume | 1 if a farmer alters legume-barley rotation | 85.5 | | | | |
| new-crop | 1 if a farmer introduced new crops | 58.9 | | | | |
| **Perceived climate change and variability** | | | | | | |
| rainfall | 1 if *meher* rainfall has been increasing for the last 25 years | 16.2 | | | | |
| temp | 1 if *meher* temperature has been increasing for the last 25 years | 90.2 | | | | |
| productivity | 1 if *meher* crop productivity has been increasing for the last 25 years | 58.3 | | | | |
| shock2 | Sum of all shock damages observed in the last 5 years | | 7.2 | 3.6 | 0 | 14 |
| **Land fragmentation indicators** | | | | | | |
| homes | Number of separate homes | | 1.5 | 0.7 | 1 | 4 |
| soil | Number of soil types an owner cultivates | | 3.9 | 1.2 | 1 | 5 |
| plots | Total number of plots per farm | | 20.6 | 13.8 | 1 | 80 |
| farm | Total farm size, ha | | 1.7 | 1.7 | 0.1 | 15.3 |
| sfi | Simpson index for LF | | 0.85 | 0.1 | 0 | 0.97 |
| distance2 | Sum of non-overlapping distance from home to plots (km) | | 2.6 | 1.6 | 0.1 | 8.7 |
| aez | 1 if a farmer cultivates both in midland and highlands | 35.7 | | | | |
| landqual | Land quality index | | 0.4 | 0.2 | 0 | 1 |

**Table 1.** *Cont.*

| Variable | Description | % | Mean | SD | Min | Max |
|---|---|---|---|---|---|---|
| **Socioeconomic characteristics** | | | | | | |
| gender | 1 if gender of household head is male | 90.6 | | | | |
| literacy | 1 if household head can read and write | 21.6 | | | | |
| experience | Farming experience (years) | | 33.9 | 15.6 | 0 | 76 |
| labour | Family labour size | | 3.5 | 2.3 | 0 | 15 |
| tlu | Number of tropical livestock units | | 2.5 | 2.7 | 0 | 22.9 |
| asset | Number of assets | | 3.6 | 2.2 | 0 | 17 |
| remittance | 1 if a household has remittance | 11.5 | | | | |
| income | Amount of off-farm income (thousands) | | 1.6 | 2.7 | 0 | 20 |
| **Social network, market access and extension services** | | | | | | |
| network1 | Participation in 5 social networks per month | | 5.8 | 5.3 | 0 | 54 |
| network2 | No. of trusted social networks | | 2.2 | 1.4 | 0 | 5 |
| credit | 1 if access to credit without constraint | 58.9 | | | | |
| market | Walking distance to market (hour) | | 1.9 | 1.2 | 0.3 | 6 |
| training1 | 1 if training on modern input use | 82.5 | | | | |
| training2 | 1 if training on soil fertility | 81.5 | | | | |
| training3 | 1 if training on soil erosion control | 84.5 | | | | |
| | Number of observations | | | | | 297 |

Focus group discussions with farmers and experts revealed that farmers perceived precipitation to be decreasing and more variable. The farmers reported rain shortages and unseasonable rain to be prevalent, resulting in low productivity and destruction of harvests. They also mentioned that temperatures have been rising in the last 25 years in both cropping seasons. Moreover, because of the changing climate, crops endemic to the midlands are currently also being grown in highland areas and the productivity of some trees (e.g., bamboo) and crops is decreasing.

This study hypothesises that farmers respond to perceived climate change variability and impacts in order to cope with hazards and take advantage of opportunities. The frequent observation of shocks (both climatic and non-climatic) by farmers in the last five years is expected to favor adaptation. Almost 95% of farmers reported adjusting their land management practices in response to increased perceived climate variability and change, while others reported practising autonomous adaptation through the continued use of customary practices.

### 3.1.2. SLM Practices

A total of 13 SLM measures potentially used as climate adaptation measure was surveyed. A chi-square test was conducted to test whether these measures could de facto be associated with perceived climate change and variability. SLM measures that were correlated with at least one indicator of climate change or variability perceived by farmers at less than 10% significance level were considered as climate adaptation measures in the study. Some SLM practices, such as conservation tillage and changing the sowing period, were not significantly correlated with farmers' perception. Eucalyptus tree planting and fertilizer use to adapt to climate change and variability were also not considered in the MVP analysis, as the government forces farmers to apply fertilizer and, hence, only eight farmers reported not using fertilizer. Eucalyptus trees are replacing indigenous trees and are not eco-friendly.

Finally, eight SLM measures were considered as adaptation practices: (i) expansion of manure use on a plot, (ii) use of a quality seed, (iii) introduction of a new crop, (iv) plant more enset (*Ensete ventricosum*), a perennial crop in the banana family, (v) plant indigenous trees, (vi) barley-legume rotation, (vii) crop diversification, and (viii) terracing (i.e., use of stone bunds or terrace steps on hilly plots). The use of these SLM practices ranged between 59% and 97% (Table 1).

### 3.1.3. LF Indicators

Seven LF indicators were used: number of plots and number of soil types across the non-contiguous plots, an agro-ecological zone dummy for growing crops in either a single

agro-ecological zone or both the midlands and highlands, sum of non-overlapping distance from the homestead to owner's plots in km, Simpson index, total farm size, and number of separate homes (see Table 1). The non-overlapping distance to plots from the homestead was measured using GPS to avoid the overlap of distances. Where the owner had multiple homes in separate plots, the distance was measured from the closest homestead to each plot.

The Simpson index, which was used to measure LF, is defined as $[1 - (\sum_{i=1}^{n} a_i^2 / A^2)]$, where $a_i$ is the area of the $i$th plot in hectares, and $A$ is the farm size in hectares, which equals the sum of the area of all $n$ plots of the farm, $A = \sum_{i=1}^{n} a_i$. The value of the index ranges from zero to one. A value of zero refers to the situation where a household has a single parcel, i.e., perfect land consolidation, whereas a value close to one implies that a landholder has many plots and the farm is severely fragmented [47]. Ownership of multiple homes is a customary practice in Gamo communities to avoid commuting costs, and is expected to enhance the use of adaptation measures. Land quality is measured as the ratio of the number of plots with vigorous barley growth to the sum of all plots that have stunted and vigorous barley growth, and quality land is assumed to increase adaptation decisions.

The level of LF in the Gamo Highlands is substantially higher than the national average landholding of 1.01 hectares fragmented into 2.4 plots [48]. In our sample, the mean farm size of households was 1.7 hectares and was fragmented into 21 parcels (Table 1). Farm-level LF varied significantly, ranging from a single consolidated plot to a maximum of 80 plots per farm. The average Simpson index was 0.85 and ranges from zero, perfect consolidation, to 0.97. The mean non-overlapping distance between parcels and the homestead was 2.6 km. We argue that higher LF is expected to decrease the use of most SLM practices.

### 3.1.4. Socioeconomic Characteristics of Households

These variables include gender, the farming experience and education of the household head, family labor size (i.e., family members aged between 14 and 65 years), off-farm income, and wealth indicators such as assets, farm implements and the number of tropical livestock units (TLUs). The number of TLUs is calculated as the weighted (TLU = $0.5 \times$ calf + $1 \times$ cow + $1.1 \times$ ox + $0.80 \times$ horse + $0.5 \times$ donkey + $0.7 \times$ mule + $0.1 \times$ sheep + $0.1 \times$ goat + $0.01 \times$ chicken) sum of individual livestock units owned. Off-farm employment opportunities were inaccessible to 72%. Farmers who had access to off-farm employment earned a mean annual off-farm income of 700 birr (approximately €28), making access to land essential for survival. We hypothesize that either wealth or human capital indicators will promote the use of adaptation measures.

### 3.1.5. Social Capital, Market Access and Extension Services

We considered two forms of social capital: (i) social network (defined as the average number of household members participating in five important types of local networks, namely, 'idir' (informal insurance to support each other during misfortune and is financed by members contribution), 'iquib' (a saving and credit association that operates by pooling money from members), 'mahiber' (a religious or cultural interest group that has parties/social gatherings and whose members help each other when needed), 'debo' (a group of people who work together to achieve common goals), and farmers' clubs (a group of five farmers organised by local government to exchange good farming practices) per month) and (ii) perceived support (which is the number of these networks a household member participates in and trusts to support him in case of misfortune). These social networks are expected to have either a positive or negative effect on the use of adaptation measures. Market access refers to the walking distance to the market in hours. Limited market access results in a lengthy mean round trip of 3.9 h for the surveyed farmers. Goods were expensive in these markets, hence, 41% of farmers said that they abstained from making purchases. Sufficient access to market is assumed to increase the use of adaptation measures. Access to credit is defined as unconstrained access to credit by farmers when they need it. Credit per annum was accessible to most farmers (59%) when required. Access to microcredit was limited to 13% with a mean loan size of 510 birr (approximately €20) in

the last five years. More common was informal credit with an average loan of €18 and accessibility of 57% per year. Loan access was mainly constrained by the fear of being unable to repay the loan (29%) and high interest rates (13%). Unconstrained credit access is expected to improve the use of adaptation measures.

### 3.2. Factors Affecting the Use of SLM Practices

Table 2 presents the estimation results. The MVP model fits the data well because the Wald test of the null hypothesis that all regression coefficients in each equation are jointly equal to zero was rejected at the 1% significance level. The MVP model is also superior to the binary probit model as the likelihood ratio test rejected the absence of correlation between the error terms at the 1% significance level (see Table 2) and 15 of the 28 pairs of error terms had a significant positive correlation. For instance, indigenous tree planting was positively correlated with most measures at the 1% significance level, except for quality seed and manure use, implying that these measures are complementary.

**Table 2.** Factors affecting the use of SLM practices.

| Variables | Manure | Seed | Indigenous | Terrace | Diversifcation | Enset | Legume | New-Crop |
|---|---|---|---|---|---|---|---|---|
| **Perceived climate variability and change** | | | | | | | | |
| rainfall | 0.049 | −0.445 * | −0.460 ** | −0.115 | −0.409 | 0.624 ** | −0.342 | 0.042 |
| temp | −0.221 | −0.234 | 0.306 | 0.556 * | −0.215 | 0.346 | 0.591 ** | 0.561 ** |
| productivity | 0.470 * | 0.061 | 0.210 | 0.327 | 0.494 *** | 0.022 | 0.469 ** | 0.343 ** |
| shock2 | −0.029 | −0.028 | −0.009 | 0.034 | −0.003 | −0.107 *** | −0.020 | − 0.033 |
| **LF indicators** | | | | | | | | |
| plot | 0.028 ** | 0.010 | 0.015 | 0.033 *** | 0.035 ** | 0.020 ** | −0.004 | 0.026 *** |
| soil | 0.174 * | 0.065 | 0.135 * | 0.206 ** | 0.244 *** | 0.130 | 0.011 | 0.071 |
| distance2 | −0.027 | −0.115 | −0.069 | −0.128 | −0.116 | 0.041 | 0.067 | −0.004 |
| sfi | −0.947 | −0.810 | −1.278 | −1.366 | 0.461 | 0.573 | 2.969 *** | −2.107 ** |
| aez | −0.241 | 0.043 | 0.013 | 0.480 * | −0.241 | −0.037 | −0.685 0.685 *** | −0.134 |
| homes | −0.219 | −0.121 | −0.157 | −0.062 | −0.107 | −0.084 | 0.234 | 0.087 |
| land2 | −0.086 | −0.168 ** | −0.109 | −0.048 | −0.036 | −0.014 | 0.172 | −0.034 |
| landqual | 0.868 | −0.171 | 0.205 | 1.850 *** | 0.233 | 0.337 | −0.095 | −0.006 |
| **Socioeconomic characteristics** | | | | | | | | |
| gender | −0.703 | 0.019 | 0.119 | 0.161 | 0.522 * | −0.367 | −0.480 | 0.396 |
| literacy | | 0.517 * | 0.339 | 0.400 | 0.254 | −0.152 | 0.080 | 0.481 ** |
| experience | −0.025 *** | −0.013 ** | −0.010 * | 0.000 | −0.005 | 0.001 | 0.005 | −0.001 |
| tlu | 0.016 | 0.028 | 0.149 ** | −0.016 | 0.028 | −0.094 ** | −0.034 | −0.078 * |
| labour | −0.033 | 0.043 | −0.052 | −0.079 ** | −0.011 | −0.046 | −0.031 | 0.045 |
| income | 0.079 | −0.251 | −0.101 | 0.139 | 0.110 | 0.246 | −0.614 * | 0.069 |
| remittance | 0.019 | 0.224 *** | 0.030 | −0.039 | 0.123 * | 0.035 | 0.241 *** | 0.043 |
| asset | −0.703 | 0.019 | 0.119 | 0.161 | 0.522 * | −0.367 | −0.480 | 0.396 |
| **Social network, market access and extension services** | | | | | | | | |
| network1 | 0.033 | 0.002 | 0.017 | 0.026 | 0.057 | 0.058 | −0.024 | −0.023 |
| network2 | −0.196 | 0.107 | 0.071 | 0.032 | −0.187 * | 0.053 | −0.020 | 0.234 *** |
| credit | 0.215 | −0.234 | 0.100 | −0.180 | −0.050 | −0.169 | −0.550 ** | −0.119 |
| market | 0.186 | 0.054 | −0.186 ** | 0.049 | −0.050 | −0.017 | 0.009 | 0.037 |
| training1 | | 0.595 *** | | | | | | |
| training2 | 1.023 *** | | | | | 0.216 | 0.632 ** | −0.121 |
| training3 | | | 0.691 *** | 0.748 *** | | | | |
| constant | 2.293 ** | 0.787 | 0.413 | −1.215 | −1.070 | −0.287 | −1.751 * | −0.329 |

| Wald statistics | $\chi^2(198) = 2427.14\ p > \chi^2 = 0.000$ |
|---|---|
| likelihood ratio test | $\rho_{21} = \rho_{31}, \ldots = \rho_{87} = 0 : \chi^2(28) = 127.2\ p > \chi^2 = 0.000$ |
| log likelihood | −824.2 |
| number of observations | 297 |

### 3.2.1. The Role of Climate Change and Variation

Farmers responded to their perception of climate variability and climate change impacts. For example, the likelihood ratio test indicates that three climate change and variability indicators jointly affected the use of SLM practices at the 1% significance level. More specifically, increases in rainfall were likely to decrease the use of certain adaptation measures, such as the use of quality

seed and planting of indigenous trees, but increase enset planting. The first group of adaptation practices are mostly perceived as being able to endure shortages of rainfall, whereas enset is more productive when rainfall is high. Hence, by planting more enset, farmers aim to take advantage of this perceived opportunity. Farmers who perceived increased temperature were more likely to increase legume-barley rotation and new crop introduction, probably as higher temperature enables growing a wider range of crops. Moreover, anticipating increased crop productivity is generally a strong incentive for applying adaptation measures. The author of [49] found that farmers switch crops to adjust to their local climatic conditions, and pastoralists in southern Ethiopia are changing their choice of adaptation practices to adapt to rising temperature [50]. However, the author of [51] found no clear effect of rising temperatures and precipitation on the use of various agricultural adaptation practices.

### 3.2.2. The Role of LF Indicators

Fragmentation indicators jointly had a significant effect on the use of SLM practices at the 1% significance level. Contrary to expectations, Table 2 shows that most LF indicators had positive effects on the use of various SLM measures. For instance, higher fragmentation is expected to inhibit terracing, but some LF indicators, such as the production of crops in both the midlands and highlands, cultivation of a higher number of plots, and different soil types were found to increase terracing. This is probably because the terracing of the plots is imperative to prevent rainfall run-off from sloping land by diminishing the degree of slope in order to maintain at least status quo production and consumption. Farmers commonly construct terraces from the soil itself, rather than from stone, due to stone and labor shortages. Unlike constructing a stone terrace, gradual terracing through soil bunds is not labor intensive; hence, farmers continue to terrace their fields (which are fragmented). Moreover, a higher number of plots and different soil types cultivated were found to be positively associated with crop diversification, probably to fit the different soil types and stabilize production.

Additionally, the cultivation of a larger number of plots and soil types was found to increase manure application and the cultivation of a larger number of plots also increased planting of perennials, although these practices require much effort. Three-quarters of farmers said that fragmentation posed a challenge when applying manure. However, these econometric results are in contrast to the results of a study by the author of [20] in northern Ethiopia. The contrasting results are perhaps associated with the customary practices of Gamo communities, which aim to ease the application of demanding land management practices. For instance, to overcome the challenges involved in carrying and applying manure to remote fields, 57% of farmers raised horses or mules to transport the manure and 45% constructed multiple homes in separate plots to stock manure to simplify application in remote plots. Moreover, traditionally, operators' homes are ringed by enset. A Spearman correlation test shows that the construction of multiple separate huts and the number of horses owned had a significant positive correlation with the number of plots on which sufficient manure was applied. Hence, in this study, only 29% of farmers reported abstaining from manure application because their plots were too fragmented.

Terracing by farmers across hillsides was not labor intensive; as a result, production in multiple agro-ecological zones promoted terracing, but unexpectedly decreased legume-barley rotation. Half of smallholders contended that plot dispersion restricted the cultivation of legumes to increase soil fertility (see Table 3). Farmers who live in midland areas, but own plots both in the midlands and the highlands decreased legume-barley rotation as highland plots suitable for legume production are far from home and the crops were being stolen. A larger number of plots increased the introduction of new crops, perhaps because having many plots tends to increase access to different soil types and microclimates suitable for new crops. However, a higher Simpson index discouraged the introduction of new crops.

**Table 3.** Land fragmentation constraints to the adoption of SLM practices (Note: Possible benefits of LF like risk reduction are not reported here).

| Selected SLM Practices (n = 279) | Yes | % |
|---|---|---|
| Does land fragmentation increase loss of your labor time by increasing the commuting time? | 179 | 60 |
| Does land fragmentation impede manure application? | 223 | 75 |
| Does land fragmentation prohibit you from planting a crop you want? | 174 | 59 |
| Does land fragmentation impede use of legume-barley rotation? | 149 | 50 |
| Does land fragmentation impede use of crop rotation? | 159 | 54 |
| Have you abstained from manure application because some of your plots are too small? | 87 | 29 |
| Is land fragmentation a challenge to apply terracing on your plots? | 135 | 46 |
| Does land fragmentation impede use of modern inputs (fertilizer and quality seeds)? | 162 | 55 |
| Does land fragmentation decrease indigenous tree planting as you want to avoid conflicts with neighboring farmers? | 138 | 47 |
| Does land fragmentation decrease eucalyptus tree planting as you want to avoid conflicts with adjacent farmers? | 175 | 59 |

Non-overlapping distance to plots had no significant effect on SLM practices used as climate adaptation strategies. However, the authors of [16,40] found a negative association between distance and investment in land improvement, while distance has a two-way effect on SLM practices according to [21]. However, farmers opined that distance-induced investment and productivity gaps between plots close to homesteads and remote plots were significant. For instance, 79% of landholders judged homestead plots to be more fertile and most producers often fallow faraway plots. Three-quarters (75%) and 90% of producers, respectively, applied more manure and planted more enset on nearby plots to improve soil fertility. Farmers (61%) perceived that erosion was more frequent on faraway plots and they terraced infields more (57%). Farmers (87%) also planted vegetables on nearby plots, and over half of them planted trees on homestead plots; 73% of farmers said they visited nearby plots more than distant ones. More frequent plot visits increase the chance of detecting land management problems (e.g., blocked drainage channels). Experts and extension agents also agreed that farmers tend to invest more in nearby plots to minimize effort and avoid wasting time. As a result, infields were more fertile and productive than faraway plots. For instance, barley yield per hectare was higher in proximate plots for the majority of the producers (58%). Farmers said remote plots were commonly share-cropped or chemical fertilizers were applied instead of manure, which may result in relatively low soil fertility. These fertility and productivity differences translated into investment differentials. Other studies have also found that farmers often cultivate remote plots less intensively; the authors of [19,52] found that rice output and labor input decrease with the distance of a plot from the homestead.

The implication is that LF increases the distance to multiple plots, resulting in loss of labor time and limiting the effective use of SLM practices as adaptation measures. The majority of farmers (60%) asserted that having scattered plots increases their commuting time, resulting in loss of labor time that can be allocated elsewhere. Labour is an important resource, particularly for the poor, to earn income and to be able to finance adaptation strategies. For example, farmers who applied quality seed have a higher off-farm income than farmers who did not. The authors of [26,53] also reported on the costly commuting consequences of fragmentation.

A remote plot may in theory have as many alternative uses as a nearby plot, but in practice remote fields tend to be at higher elevations and underutilised, implying that LF constrains the adoption of available adaptation measures. For instance, in the Gamo Highlands, 87% of producers exclusively used nearby plots to grow staple vegetables (e.g., potatoes and cabbages). Vegetables are bulky to transport, labor intensive to cultivate, fence, and manure, and outlive other crops, increasing their exposure to livestock attacks and theft in open outfields without fences; this constrains their production on remote plots. Likewise for Mennonite farmers in Canada, their cropping patterns are constrained by land dispersion, so they prefer infields for growing heavy crops (such as potatoes), but light crops (such as wheat) are grown both in infields and outfields [23].

Large farm size decreased the use of quality seed, implying that land scarcity can induce agricultural intensification in the form of the use of quality seed [41] and plot size has inconsistent effect on the adoption of SLM measures [21]. Large farm size also decreased the planting of indigenous

trees. This is not confirmed by the author of [54], who observed that large farm size increases tree planting. However, 47% of the farmers in the present study abstained from planting aboriginal trees to avoid conflict (see Table 3). The contrast between inferential and descriptive results may be attributed to the planting of eucalyptus trees, which compete for limited land and are preferred over local trees because they grow faster and have higher economic benefits as farmers opined. Despite its externality effects, farmers planted eucalyptuses more than twice as frequently as native trees, such as bamboo. Planting combinations of eucalyptus and local trees is impossible as eucalyptus hinders the growth of crops and local trees. Moreover, effects of eucalyptus trees continue for some years even after they are harvested.

### 3.2.3. The Role of Socioeconomic Characteristics

Household head literacy was found to be significant in the choice to use quality seeds and introduce new crops. Previous studies have also found that better education improves access to information on improved technologies and climate variation, and enhances adoption of adaptation measures [14,40]. The farming experience of the household head was inversely correlated with the use of manure and quality seed and native tree planting. Older farmers are unable to manage huts in separate parcels to stock manure; they also lack the money to buy quality seed. Moreover, older farmers are less likely to invest in trees that have a long payback period. In the literature, the effect of farming experience is controversial, with instances of experience discouraging the adoption of climate change adaptation measures [21,55,56] and instances where age increases adoption [51].

The number of assets and livestock owned are wealth indicators and enhance the use of legume-barley rotation and quality seed, as well as the planting of indigenous trees, but had contrasting effects on the planting of enset. However, in a study by the author of [8], better-off families have a significantly enlarged choice of measures for offsetting climate risks. In addition, a positive effect of TLU ownership was found by the author of [14].

The availability of family labor increased the likelihood of growing new crops. However, the size of off-farm income discouraged terracing as less time is left for such an adaptation measure after off-farm activities—and perhaps undertaking these activities—results in a high-opportunity cost for farmers who can earn higher income off-farm.

### 3.2.4. The Role of Social Networks, Market Access and Extension Services

Participation in networks did not have a significant effect on SLM practices, but the number of trusted social networks a farmer had increased the introduction of new crops. In the literature, social capital has both a positive and negative correlation with the adoption of sustainable intensification practices [41] and adaptation methods [9]. However, social networks are vital to the exchange of information among farmers constrained by market imperfections, as argued by the author of [54]. Moreover, we observed that the number of trusted social networks from which support is expected during misfortune discouraged crop diversification. Perceived trust in support from networks acts as a risk-avoidance strategy and inhibits the adoption of measures that hold some risk, such as introducing new crops.

Lengthy market access decreased indigenous tree planting and had no effect on the use of other measures. Studies by the author of [8] found that proximity to the output market discourages the uptake of adaptation measures. However, the author of [51] found that remoteness from markets has both desirable and undesirable effects. Access to credit decreased legume-barley rotation. However, other studies have shown instances of both positive and negative associations. Some studies found that credit discourages adaptation [9,57,58].

Access to focused extension trainings was found to enhance quality seed use, manure use and enset planting to boost soil fertility, as well as terracing and the planting of native trees for erosion control purposes. The findings of this study on the effect of training agree with those of [59,60]. More importantly, [7] demonstrates that information provided by extension workers increases the

probability of farmers deciding to adapt. In addition, [54] explains that highly-skilled extension workers enhance the adoption of sustainable agricultural packages, implying that upgrading the skills of extension workers could enhance uptake.

## 4. Concluding Remarks

Considering LF in the context of key environmental, socio-economic and perceptual factors, this study delivers integrated insights into the conditions under which farmers use SLM practices as climate adaptation strategies. We present empirical evidence that agricultural land across the Gamo Highlands in Ethiopia is highly fragmented, exceeding the national average land fragmentation levels. Land fragmentation poses a challenge to the application of effective SLM practices and hence can exacerbate land degradation and vulnerability to food insecurity and other climate change impacts in the already food-insecure Gamo Highlands.

The study found that farmers in the Gamo Highlands altered the use of SLM practices to adapt to perceived climate change and variation. MVP model results indicate that LF created both opportunities and challenges for the application of SLM practices as a means of climate change adaptation. The magnitude and direction of the effects depend on the type of index used to capture LF and the adaptation methods employed. The qualitative analysis of farmers' perceptions found that fragmentation was a challenge for adaptation by obstructing choice and effective use of available adaptation measures. For instance, in more fragmented landscapes, farmers used less SLM strategies in remote fields than in fields close to their homestead due to increased commuting costs. In addition, farmers invest in homestead plots more frequently, such that nearby plots were less eroded and more productive than remote plots. Moreover, farmers perceived nearby plots as having more alternative uses than faraway fields. Contrasts between econometric results and farmers' perceptions may be attributed to the fact that farmers construct separated multiple homes and use horses and mules for transportation to ease the application of demanding SLM practices.

This analysis demonstrates that LF is not necessarily detrimental to adaptation, as it promotes the use of some climate change adaptation practices. For instance, higher fragmentation promoted crop diversification (to fit different soil types and to stabilize yield), manure application (to increase soil fertility), and terracing (to prevent erosion on mountainous farm fields). Therefore, although LF is exogenous to farmers, it could be deployed in climate change adaptation planning. For example, the assembling of nearby small plots in different microclimates could be an important strategy to maximize the benefits of fragmentation to climate adaptation, while at the same time increasing the magnitude and quality of investment in remote fields. Land assembling can be done either by promoting the voluntary exchange of plots between farmers or through cooperatives. Moreover, increasing farmers' access to focused training in SLM practices is vital to promote adaptation.

This study examined whether LF is facilitating or obstructing the use of SLM practices to adapt to climate change and variability at the farm level. However, farmers make adaptation decisions at the plot level. At the same time, the underlying conditions for the adoption and maintenance of adaptation measures may change over time. Future studies would, therefore, be useful to explore the effects of land fragmentation in relation to the plot-level dynamics of sustainable land management and climate adaptation including the intensification, modification, abandonment and replacement of particular strategies. We also suggest further research that identifies fit-for-purpose land consolidation strategy that promotes sustainable development [61] in the study area. Studies by the authors of [61–63] could serve as a starting point for that.

**Author Contributions:** T.C.C. and L.F. conceived the research idea; T.C.C. and J.P. designed the methods; T.C.C. and D.S. wrote the draft version of the paper, which was reviewed and edited by L.F. and J.P.

**Funding:** This research was financed by the Netherlands Organization for International Cooperation in Higher Education (Nuffic), under grant no. NICHE/ETH/020, and coordinated by Tilburg University.

**Conflicts of Interest:** The authors declare no conflict of interest.

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
