# Peer review of "Is Land Fragmentation Facilitating or Obstructing Adoption of Climate Adaptation Measures in Ethiopia?"

_sustainability, doi:10.3390/su10072120_

Round 1

Reviewer 1 Report

General comments

The article investigates the reasons for fragmentation and possible land management solution strategies which are sustainable.   An econometric technique, based on household surveys, is used to assemble and aggregate farmers’ opinions and stated choices on use of land, and on the visible and observed characteristics of climate change.  Overall, the article is relies on statistical analysis, which is sufficiently convincing.   

Specific comments

Section 1.  -Land fragmentation (LF) is described as completely exogenous to farmers (i.e., it is not an adaptation strategy that they choose because sons inherit fragmented land from their parents). This is of course not entirely true. It is indeed true that fragmentation might come from inheritance, but also speculation and frequent transfer of land (use) rights may be reasons for LF. Hence, this sentence requires a bit of adaptation.  Moreover, farmers do have possibilities to address LF, notably the solutions which are the main conclusions / recommendations – i.e. ‘voluntary assembling of neighbouring plots’. This is not a new solution, but one which is already well-known and well-described – namely land consolidation.  There are different land consolidation strategies possible, including those by voluntary exchange of plots – see for example: (Louwsma et al. 2014; Demetriou et al. 2012; Bennett et al. 2015; Haldrup 2015; Louwsma et al. 2017).

Given these, one could have added/included the assembling consolidation strategy as one of the options which farmers could choose. Currently the Table 3 provides only negative choices, i.e. choices as constraints. Instead one could have included more positive options in the survey. This does not affect the results per se, but this might have generated some more alternative options.    

Bennett, R. M., F. A. Yimer, and C. Lemmen. 2015. Toward Fit-for-Purpose Land Consolidation. In Advances in Responsible Land Administration: CRC Press, 163-182.

Demetriou, D., J. Stillwell, and L. See. 2012. Land consolidation in Cyprus: why is an integrated planning and decision support system required? Land Use Policy 29 (1):131-142.

Haldrup, N. O. 2015. Agreement based land consolidation – In perspective of new modes of governance. Land Use Policy 46 (0):163-177.

Louwsma, M., C. Lemmen, M. Hartvigsen, J. Hiironen, and J. Du. 2017. Land Consolidation and Land Readjustment for Sustainable Development–the Issues to Be Addressed. In FIG Working Week 2017. Helsinki, Finland.

Louwsma, M., M. Van Beek, and B. Hoeve. 2014. A new approach: Participatory Land Consolidation XXV FIG Congress, 16-21 June 2014, Kuala Lumpur, Malaysia. FIG, 2014. , 10.

Author Response

Response to Reviewer 1 comments

First, we would like to say thank you for your valuable comments which helped us to improve our paper. Below, we addressed your comments one by one.

Part I: General comments and responses

English language and style

( ) Extensive editing of English language and style required
( ) Moderate English changes required
(x) English language and style are fine/minor spell check required
( ) I don't feel qualified to judge about the English language and style

Yes

Can   be improved

Must   be improved

Not   applicable

Does   the introduction provide sufficient background and include all relevant   references?

( )

(x)

( )

( )

Is   the research design appropriate?

(x)

( )

( )

( )

Are   the methods adequately described?

(x)

( )

( )

( )

Are   the results clearly presented?

(x)

( )

( )

( )

Are   the conclusions supported by the results?

(x)

( )

( )

( )

Comments and Suggestions for Authors

General comments

The article investigates the reasons for fragmentation and possible land management solution strategies which are sustainable.   An econometric technique, based on household surveys, is used to assemble and aggregate farmers’ opinions and stated choices on use of land, and on the visible and observed characteristics of climate change.  Overall, the article is relies on statistical analysis, which is sufficiently convincing.   

Part II: General Comments and Responses

General Comment 1, GC1  

English language and style:

Response to GC1

The reviewer suggested that the English language and style are fine or that a minor spelling check is required. We would like to inform the reviewer and editors that we have had our manuscript checked by a professional language editor who is a native English speaker.

Part II: Specific Comments and Responses

Specific Comment 1, SC1

Section 1.  -Land fragmentation (LF) is described as completely exogenous to farmers (i.e., it is not an adaptation strategy that they choose because sons inherit fragmented land from their parents). This is of course not entirely true. It is indeed true that fragmentation might come from inheritance, but also speculation and frequent transfer of land (use) rights may be reasons for LF. Hence, this sentence requires a bit of adaptation.

Response to SC1

We still would like to argue that land fragmentation is exogenous to farmers. There are three reasons for this. First, land is inherited by farmers, which is something they cannot influence. Second, land redistribution programs have not been implemented in the Gamo Highlands unlike in the North of Ethiopia (Tigray and Amhara). Finally, farmers have land use rights in Ethiopia, but they cannot lease or sell land, implying that farmers do not have opportunities to speculate and transfer land use rights. However, we cannot exclude the possibility that farmers do exchange land use rights, but we expect this to be rare; one reason could be that because of off-farm employment, some farmers want to reduce the amount of land they farm. We now, included this text on P.2, line 22-30 adapted the wording to read ‘(largely) exogenous’.

Specific Comment 2, SC2

Moreover, farmers do have possibilities to address LF, notably the solutions which are the main conclusions / recommendations – i.e. ‘voluntary assembling of neighbouring plots’. This is not a new solution, but one which is already well-known and well-described – namely land consolidation. There are different land consolidation strategies possible, including those by voluntary exchange of plots – see for example: (Louwsma et al. 2014; Demetriou et al. 2012; Bennett et al. 2015; Haldrup 2015; Louwsma et al. 2017). Given these, one could have added/included the assembling consolidation strategy as one of the options which farmers could choose.

Response to SC2

You are right, i.e., ‘voluntary assembling of neighboring plots’ is not a new strategy and there are indeed various land consolidation models that are well explored by papers suggested by you. Although ‘voluntary assembling of neighboring plots’ is not a new strategy, farmers living in Gamo highlands have rarely consolidated their land demonstrated by the large number of plots per farm (see Table. 1). Hence we suggested this approach as a future strategy for the following reasons, although ‘voluntary assembling of neighboring plots’ is not a new strategy. First, voluntarily assembling of homogeneous neighboring plots into larger plots reduces the level of fragmentation and maintains heterogeneous plots (i.e., plots that are different in terms of soil type, slope, water retention capacity and located in different microclimates and agro-ecological zones). These heterogeneous plots per farm, encourage crop diversification over agro-ecological diversities and over seasons. Moreover, to take opportunities of agro-ecological diversities farmers produce different crops during the ‘belg’- short rainy season and ‘meher’ long rainy season in Gamo highlands of Ethiopia. Second, crop diversification over seasons and across different plots reduces production risks as farmers have poor access to risk reduction strategies. For instance, farmers in Gamo Highlands do not have access to insurance for crop and livestock failure and limited access to credit and off-farm employment opportunities. Third, perfect land consolidation or a serious reduction of the number of plots per farm leads to the loss of these benefits of heterogeneous plot ownership, which is costly to subsistence farmers. Finally, land consolidation programs, although potentially beneficial, have not been implemented in the Gamo Highlands. Therefore, we recommend that land consolidation models explored by papers suggested by the reviewer can be applied to Gamo Highlands without overlooking and undermining the benefits of agro-ecological diversities for subsistence farmers. In a mini-workshop held to discuss research findings and policy implications to farmers and agricultural experts in the study area farmers suggested step-by-step land consolidation rather than perfect land consolidation at a time. For instance, farmers discussed that in the first round farmers can consolidate 5% of their plots, and in the second round, they may consolidate 10% of plots and so forth until they are left with a desired number of plots. Step-by-step voluntary plot assembling can help to address the lack of double coincidence problem that may happen if farmers could barter all the plots they have at the same time. Moreover, farmers that do not want to assemble and barter their plots at the initial stage of the program can observe the benefits obtained by farmers engaged in plot assembling and show interest to barter at the later stages. We have not researched in depth possible plot assembling strategies to Gamo Highlands, and therefore we were reluctant to discuss this issue in depth.

To address your comment, we suggested for further research that identifies cost-effective land consolidation strategies to the Gamo Highlands (see P. 13 lines 23-25). We also provided the readers and researchers with some available land consolidation strategies (i.e., these consolidation strategies are suggested by the reviewer) that can be used as a reference to identify cost-effective land consolidation stratgies to the study area.

Specific Comment 3, SC3

Currently the Table 3 provides only negative choices, i.e. choices as constraints. Instead one could have included more positive options in the survey. This does not affect the results per se, but this might have generated some more alternative options.   

Response to SC3

You are  correct. We collected data on  both risk reduction benefits of fragmentation and constraints of fragmentation to sustainable land management practices. However, we reported only the latter in table 3 and excluded the former to be focused. However, we would like to remark that by answering ‘no’ farmers can indicate that land fragmentation is not constraining adoption of sustainable land management practices. To address the comment we added a footnote for table 3 that we ignore the possible benefits of LF (p.13, foot note 7).  

 Bennett, R. M., F. A. Yimer, and C. Lemmen. 2015. Toward Fit-for-Purpose Land Consolidation. In Advances in Responsible Land Administration: CRC Press, 163-182.

Demetriou, D., J. Stillwell, and L. See. 2012. Land consolidation in Cyprus: why is an integrated planning and decision support system required? Land Use Policy 29 (1):131-142.

Haldrup, N. O. 2015. Agreement based land consolidation – In perspective of new modes of governance. Land Use Policy 46 (0):163-177.

Louwsma, M., C. Lemmen, M. Hartvigsen, J. Hiironen, and J. Du. 2017. Land Consolidation and Land Readjustment for Sustainable Development–the Issues to Be Addressed. In FIG Working Week 2017. Helsinki, Finland.

Louwsma, M., M. Van Beek, and B. Hoeve. 2014. A new approach: Participatory Land Consolidation XXV FIG Congress, 16-21 June 2014, Kuala Lumpur, Malaysia. FIG, 2014. , 10.

Reviewer 2 Report

I suggest the following:

- The literature should be updated (2017 and 2018) and should consider documents as: http://www.fao.org/fileadmin/user_upload/drought/docs/Ethiopia%20Land%20Fragmentation%20Report_FINAL%207%20feb%202012.pdf;

- I would like do see more explanations about the survey used (structure, ....);

- I suggest that the authors present the citations and references of the original sources of the model presented in the section 2.3.;

- I would like to see more explanations about the specific model used for the results in the table 2, as well more explanations about the robustness and weaknesses of these results. Please, present the usual problems related with the probit models and how these problems are here present and were solved.

Author Response

Response to Reviewer 2 comments

First, we would like to thank you for your valuable comments which helped us to improve our paper. Below, we addressed your comments one by one.

Part I: General comments and responses

English language and style

( ) Extensive editing of English language and style required
( ) Moderate English changes required
( ) English language and style are fine/minor spell check required
(x) I don't feel qualified to judge about the English language and style

Yes

Can   be improved

Must   be improved

Not   applicable

Does   the introduction provide sufficient background and include all relevant   references?

( )

( )

(x)

( )

Is   the research design appropriate?

( )

( )

(x)

( )

Are   the methods adequately described?

( )

( )

(x)

( )

Are   the results clearly presented?

( )

( )

(x)

( )

Are   the conclusions supported by the results?

( )

( )

(x)

( )

Your General Comment 1, GC1  

English language and style:

Response to GC1

We would like to inform the reviewer and editors that we have had our manuscript checked by a professional language editor who is a native English speaker.

Your General Comment 2, GC2

Does the introduction provide sufficient background and include all relevant references? The reviewer suggested ‘must be improved’:

Response to GC2:

We made some revisions to improve the introduction further. We updated the literature and revised it to provide additional background information (see, P.1, line 40-43). We revised the research problem and added literature related to sustainable land management practices and land fragmentation to better pinpoint the contribution of the study (see P. 2, Line 22).

Your General Comment 3, GC3

Is the research design appropriate? The reviewer suggested ‘must be improved’

Response to GC3:

We think the research design is appropriate. However, we read carefully the parts on the research problem, data analysis techniques and data collection and made minor changes in these sections to further improve the description. For instance, we added literature to provide additional  background information and discussed data collection in more detail  and justified further the  use of the multivariate probit model e.g. by adding some additional references (P. 3, line 30).

Your General Comment 4, GC4

Are the methods adequately described? The reviewer suggested ‘must be improved’

Response to GC4:

We used a mixed method with emphasis on a quantitative method (i.e., MVP model) that enabled  to substantiate inferential results. To justify further the quantitative method selected (i.e. MVP model), we included additional literature (see P.3, Line 30).

Your General Comment 5, GC5

Are the results clearly presented? The reviewer suggested ‘must be improved’

Response to GC5

We could not change the presentation in the result section because you were not  specific enough about the changes required. 

General Comment 6, GC6

Are the conclusions supported by the results? The reviewer suggested ‘must be improved’

Response to GC6

We formulated the conclusions based on the results of our analysis. We suggested additional  research based on the results found (see the last lines of last section).

Part II: Specific Comments and Responses

Comments and Suggestions for Authors

I suggest the following:

Your Specific Comment 1, SC1

- The literature should be updated (2017 and 2018) and should consider documents as: http://www.fao.org/fileadmin/user_upload/drought/docs/Ethiopia%20Land%20Fragmentation%20Report_FINAL%207%20feb%202012.pdf;

 Response to  SC1

We updated the review of the literature based on your comments, see table below.

Literature   revised/added

Page   (P) and Line (L) number

[1],[2]

P.2,   line 42

[3]

P.2,   lines 43

[4-6]

P.4,   line 30

[7]

P.2,   lines 9-10

[8]

P.4   line 43

[9]

p.2   line 18

We also considered the reference suggested by the reviewer to update the literature review (see P.2,  lines 9-10).

Your Specific Comment 2, SC 2

- I would like do see more explanations about the survey used (structure, ....);

Response  to SC2

We added a few additional descriptions of the survey in the revised version (see, P 3, Lines 21-24). For instance, we explained that the questionnaire is used as a tool to collect quantitative data from 297 households selected randomly from 10 strata. Moreover, we mention that we had three focus group discussions (i.e., one focus group discussion per kebele[1] and in each focus group 12 persons, including farmers and extension workers participated) and held key-informant interviews with 9 persons comprising farmers and experts. 

Your Specific Comment 3, SC3

- I suggest that the authors present the citations and references of the original sources of the model presented in the section 2.3.;

Response  to SC3

We revised and considered original sources for the model (i.e., multivariate probit model) presented in section 2.3. We discuss original sources such as [4-6] that explain the multivariate probit model estimation.

Your Specific Comment 4, SC4

- I would like to see more explanations about the specific model used for the results in the table 2, as well more explanations about the robustness and weaknesses of these results. Please, present the usual problems related with the probit models and how these problems are here present and were solved.

Response to SC4

The multivariate probit model is an extension of the binary probit model that allows for more than one equation, and with correlated error terms of the different equations. We explain under which conditions the MVP model is appropriate. For instance, according to [4, 5, 8] a bivariate probit model allows the two error terms to be correlated while a multivariate probit model is appropriate  when multiple error terms are correlated. When error terms are not correlated both the bivariate and multivariate probit models are reduced to independent probit equations. However, estimating independent probit regressions gives biased and inefficient coefficient estimates when the disturbance terms are correlated [8]. This correlation is ignored in studies that use independent probit models. Your comment was useful to strengthen the paper.

Submission Date

14 May 2018

Date of this review

25 May 2018 18:22:02

1.  Kassie, M; Pender, J; Yesuf, M; Kohlin, G; Bluffstone, R;Mulugeta, E. Impact of soil conservation on crop production in the Northern Ethiopian Highlands; Intl Food Policy Res Inst, 2007.

2.   Bryan, E; Deressa, TT; Gbetibouo, AG;Ringler, C. Adaptation to climate change in Ethiopia and South Africa: Options and constraints. Environmental Science and Policy. 2009, 12, 413-426. https://doi.org/10.1016/j.envsci.2008.11.002.

3.  Deressa, TT; Hassan, RM; Ringler, C; Alemu, T;Yesuf, M. Determinants of farmers’ choice of adaptation methods to climate change in the Nile Basin of Ethiopia. Global Environmental Change. 2009, 19, 248-255. https://doi.org/10.1016/j.gloenvcha.2009.01.002.

4. Greene, WH. Econometric analysis, 5th; Pearson Education, Inc., Upper Saddle River, New Jersey, 07458: USA, 2003, 0-13-066189-9.

5.   Cameron, AC;Trivedi, PK. Microeconometrics: methods and applications; Cambridge University Press: UK, 2005, 9780521848053.

6.  Mulwa, C; Marenya, P;Kassie, M. Response to climate risks among smallholder farmers in Malawi: A multivariate probit assessment of the role of information, household demographics, and farm characteristics. Climate Risk Management. 2017, 16, 208-221.

7.   Flintan, F; Tache, B;Eid, A. Rangeland fragmentation in traditional grazing areas and its impact on drought resilience of pastoral communities: Lessons from Borana, Oromia and Harshin, Somali Regional States, Ethiopia. Oxfam: Oxford, UK. 2011.

8.   Dorfman, JH. Modeling multiple adoption decisions in a joint framework. American Journal of Agricultural Economics. 1996, 78, 547-557. https://doi.org/10.2307/1243273

9.  Sklenicka, P;Salek, M. Ownership and soil quality as sources of agricultural land fragmentation in highly fragmented ownership patterns. Landscape ecology. 2008, 23, 299-311.

[1] Kebele is the lowest administrative unit in Ethiopia.

Round 2

Reviewer 2 Report

The authors may improve the explanations about the model and the results.

Author Response

The authors may improve the explanations about the model and the results.

Response

We made minor change to address the concerns of the reviewer, compared to the previous round we added some references, e.g., [1, 2], that further explain the model (see p. 3 line 30 and p. 4 line 8).